# Role of Genetic Thrombophilia Markers in Thrombosis Events in Elderly Patients with COVID-19

**DOI:** 10.3390/genes14030644

**Published:** 2023-03-04

**Authors:** Irina Fevraleva, Daria Mamchich, Dmitriy Vinogradov, Yulia Chabaeva, Sergey Kulikov, Tatiana Makarik, Vahe Margaryan, Georgiy Manasyan, Veronika Novikova, Svetlana Rachina, Georgiy Melkonyan, Karine Lytkina

**Affiliations:** 1National Medical Research Center for Hematology, Novy Zykovski Lane 4a, 125167 Moscow, Russia; 2Hospital Therapy Department No. 2, I.M. Sechenov First Moscow State Medical University, RF Health Ministry, Bolshaya Pirogovskaya St. 2, Bld. 4, 119435 Moscow, Russia; 3War Veterans Hospital N3, Startovaya St. 4, 129336 Moscow, Russia

**Keywords:** inherited thrombophilia, thrombosis, real-time AS-PCR, hereditary mutation

## Abstract

Thrombosis is an extremely dangerous complication in elderly patients with COVID-19. Since the first months of the pandemic, anticoagulants have been mandatory in treatment protocols for patients with COVID-19, unless there are serious contraindications. We set out to discover if genetic thrombophilia factors continue to play a triggering role in the occurrence of thrombosis in patients with COVID-19 with prophylactic or therapeutic anticoagulants. We considered the following genetic markers as risk factors for thrombophilia: G1691A in the *FV* gene, C677T and A1298C in the *MTHFR* gene, G20210A and C494T in the *FII* gene, and (−675) 4G/5G in the *PAI-I* gene. In a cohort of 176 patients, we did not obtain a reliable result indicating a higher risk of thrombotic complications when taking therapeutic doses of anticoagulants in carriers of genetic markers for thrombophilia except the C494T mutation in the *FII* gene. However, there was still a pronounced tendency to a higher incidence of thrombosis in patients with markers of hereditary thrombophilia, such as *FV* G1691A and *FII* G20210A mutations. The presence of the C494T (Thr165Met) allele in the *FII* gene in this group of patients showed a statistically significant effect of the mutation on the risk of thrombotic complications despite anticoagulant therapy.

## 1. Introduction

Coronavirus infection is a systemic disease and has a wide range of clinical manifestations. In addition to affecting the respiratory system, severe acute respiratory syndrome coronavirus 2 (SARS-CoV-2) affects the cardiovascular, nervous, digestive, and other systems [1,2,3]. While some patients had mild disease, a large proportion of patients developed severe complications such as acute respiratory syndrome or disseminated intravascular coagulation with subsequent sepsis, multiple organ failure, and death within a short time after infection. Thrombosis is a very serious and frequent complication of Coronavirus Disease 2019 (COVID-19). The incidence of such events as venous thrombosis, ischemic myocardial infarction, ischemic stroke, and pulmonary embolism (PE) in COVID-19 patients in intensive care units has been shown to be 16.5–50.0% [4,5]. Since the first months of the pandemic, anticoagulants have been mandatory in treatment protocols for patients with COVID-19, unless there are serious contraindications [6].

The factors leading to a high risk of trombotic complications in patients with COVID-19 can be divided into acquired and hereditary factors. Acquired factors are old age, obesity, and comorbidities, especially diabetes, arterial hypertension, and antiphospholipid syndrome [7,8,9,10]. Other factors and direct viral effects include increased vasoconstrictor angiotensin II, decreased vasodilator angiotensin and sepsis-induced cytokine release. All of these preconditions can cause coagulopathy in COVID-19. COVID-19 is especially dangerous for elderly patients who are more prone to develop severe complications of COVID-19 infection and have a higher risk of thrombosis.

Furthermore, there are many published works proving that people carrying G1691A mutations in the proaccelerin gene (*FV*, Leiden mutation), G20210A in the prothrombin *FII* gene, (-675) 4G/5G in the plasminogen activator inhibitor (*PAI-I*) gene, C677T, A1298C in the methylenetetrahydrofolate reductase gene (*MTHFR*), etc., have a higher risk of thrombosis [11,12,13]. It would be logical to assume that patients with COVID-19 having genetic predisposition factors for thrombophilia are more likely to have thrombosis than those who do not have such mutations. However, few studies on the thrombosis events and anticoagulant therapy in patients with thrombophilia markers have been reported so far [14,15,16]. Do genetic thrombophilia factors in patients with COVID-19 continue to play a triggering role in the occurrence of thrombosis with prophylactic or therapeutic anticoagulants? The answer to this question is the task of our study.

## 2. Materials and Methods

### 2.1. Study Cohort

The work was carried out as part of a prospective observational study in a multidisciplinary geriatric hospital in Moscow, Russia. All participants gave their written informed consent to participate in the study. The study included 176 patients aged 50 to 100 years from among those hospitalized with a diagnosis of coronavirus infection between November 2021 and January 2022. The diagnosis was confirmed by PCR or SARS-CoV-2 antigen rapid test.

### 2.2. Target Parameters and Risk Factors

The severity of COVID-19 disease in patients was determined by some parameters such as the volume of lung damage according to computed tomography (CT), whether the patient needs to be transferred to the intensive care unit (ICU), and the outcome of hospitalization—recovery, death, or transfer to another hospital. We compared these parameters with the results of genetic analysis of patients’ DNA for the presence of mutations that are markers of thrombophilia. We analyzed clinical thrombotic complications in elderly COVID-19 patients, such as the presence of venous thromboembolism according to ultrasound vascular Doppler sonography (VTE), the presence of ischemic acute cerebrovascular injury (ACE), pulmonary embolism (PE), or myocardial infarction (MI) during hospitalization, and their association with genetic markers of thrombophilia, using data from patients’ medical records.

### 2.3. Genetic Analysis

DNA was isolated from buccal epithelial cells of the patients using a Ribo-Prep kit (AmpliSens, Russia) for total RNA/DNA isolation from clinical material. Subsequent analysis of genetic mutations in genomic DNA was performed by allele-specific polymerase chain reaction (AS-PCR) on a CFX96 Touch Real-Time PCR Detection System (Bio-Rad Laboratories Inc., USA). The genes under study, mutation localization, primers, Taqman assays, and AS-PCR conditions are listed in Table 1.

PCR conditions for the Taqman real-time AS-PCR were preheating to 94 °C, 300 s; the next 45 cycles were thermal cycling: denaturation: −94 °C for 20 s, annealing, and elongation: −60 °C for 50 s. Each primer quantity was 10 pM/per reaction; each probe quantity was 5 pM/per reaction; reaction volume was 25 μL. PCR buffer, MgCL2, dNTP, and Taq-polymerase were provided by LLC Syntol (Moscow, Russia). All primers and Taqman probes were synthesized by LLC Syntol (Moscow, Russia) according to our author’s design.

We used multiplex real-time allele-specific polymerase chain reaction in Taqman format to detect the six genetic thrombophilia markers: G1691A in the *FV* gene, C677T and A1298C in the *MTHFR* gene, G20210A and C494T in the *FII* gene, (-675) 4G/5G in the *PAI-I* gene. We developed a system for real-time AS-PCR consisting of six sets of primers and probes labeled with fluorochromes and their quenchers. For each polymorphism, a pair of primers were chosen so that the last nucleotide at the 3’ end of the first primer matched the normal nucleotide in the gene sequence, and the last nucleotide at the 3’ end of the second primer matched the mutant nucleotide in the gene sequence (Table 1). In the Table these primers are labeled with “w” and “mt”. For each pair, the reverse common primer was selected so that the PCR amplicon length was 150–250 nucleotide pairs. In addition, a Taqman probe labeled with one of the four fluorochromes and an appropriate quencher was selected. For multiplex AS-PCR, 4 types of PCR mix were prepared in four tubes, differing in the set of primers. Using a mixture of primers and probes detecting normal alleles of the *FV* (G1691), *FII* (G20210), *MTHFR* (C677), and *PAI-I* (5G) genes, we thus performed four independent PCR reactions in the first tube. The first mix contained primers to detect normal alleles of the *FV* (G1691), *FII* (G20210), *MTHFR* (C677), and *PAI-I* (5G) genes. This approach allowed us to simultaneously analyze the presence or absence of the indicated normal alleles in the *FV*, *FII*, *MTHFR*, and *PAI-I* genes. The second mix contained primers to detect mutant alleles of the *FV* (1691A), *FII* (20210A), *MTHFR* (677T), and *PAI-I*(4G) genes. Similarly, the presence or absence of mutant alleles of the *FV* (1691A), *FII* (20210A), *MTHFR* (677T), and *PAI-I* (4G) genes was identified in the second tube. The third tube included primers to detect normal alleles of the FII (C494) and *MTHFR*(A1298) genes and the fourth included primers to detect mutant alleles of *FII* (494T) and *MTHFR* (1298C) genes.

### 2.4. Statistical Analysis

Standard methods of descriptive statistical analysis were used in the work. To assess the association of target parameters with the risk factors, methods of frequency analysis were used. The chi-square test was used to assess the significance of the association. The odds ratio with a corresponding 95% confidence interval was given as a measure of association. To test hypotheses about the presence of differences in the distributions of numerical parameters in compared groups after testing the hypothesis of normality of the distribution, a *t*-test was used

## 3. Results

### 3.1. Initial Characteristics

Our study included 176 patients with the average age of 73 (50–100) years, with the prevalence of women (63.6%). Most patients had comorbidities, the most common being hypertension (HD) 148 (84.6%), coronary heart disease (CHD) 74 (41.3%), and diabetes mellitus (DM) 59 (33.5%) in the study subjects. The duration of hospitalization was 3–66 days. Hospital mortality was 46 (26.1%). Patients were prospectively investigated for in-hospital thrombotic complications. The initial demographic and clinical characteristics of the patients are presented in Table 2, Table 3 and Table 4.

### 3.2. Thrombotic Complications

The cohort of elderly patients treated for COVID-19 at our hospital was divided into two groups. The first group consisted of patients who had such thrombotic complications as VTE, thrombosis according to ultrasound investigation, ischemic AMI, and myocardial infarction; the second group consisted of those who did not demonstrate thrombosis complications during the disease. There were 132 patients in the group without complications, and 44 patients in the group with solitary thrombotic complications, where 5 of them suffered from combined thrombosis (Table 3).

The results presented in Table 3 show that patients with thrombosis are more than 1.5 times as often admitted to the ICU, and their hospital mortality is 2.5 times higher.

The genetic markers of thrombophilia in our study were heterozygous Leiden mutations in the *FV* gene and G20210A in the *FII* gene (corresponding homozygous mutations were not found in patients); homozygous C677T and A1298C mutations in the *MTHFR* gene and 4G/5G in the *PAI-I* gene; and hetero- and homozygous forms of C494T mutation. We considered the absence of Leiden, G20210A, and C494T mutations as well as normal and heterozygous C677T and A1298C genotypes in the *MTHFR* gene and 4G/5G in the *PAI-I* gene as the absence of an appropriate marker of thrombophilia.

In our study cohort of 176 elderly patients, the Leiden mutation in the *FV* gene was detected in 8 (4.6%) patients, the G20210A mutation in the *FII* gene was detected in 3 (1.7%) patients, and at least one mutation in the C494G allele of the *FII* gene was detected in 55 (31.1%) patients. Homozygous C677T and A1298C mutations in the *MTHFR* gene were observed in 20 (11.36%) patients; 57 (32.4%) patients had a homozygous 4G/5G mutation in the *PAI-I* gene. Table 4 shows the prevalence of the studied mutations in the groups of patients with and without thrombotic complications. Genetic thrombophilia markers are highlighted with gray in this table and below.

Statistical analysis showed that *p*-criterion of significance does not exceed 0.05 only in the case of the *FII* gene C494T mutation, i.e., of all the mutations studied, only the presence of at least one C494T allele significantly increases the risk of thrombotic complications in elderly COVID-19 patients. As for *FV* G1691A and *FII* G20210A mutations, we observed a definite trend toward an increased probability of thrombosis in patients with these mutations relative to patients with the mentioned mutations without thrombosis (9.1 versus 3.0 and 4.5 versus 0.8 at *p* = 0.09, respectively). Clearly, because these mutations are rare, a large cohort of patients is required for reliable results. We found no association between the presence of homozygous mutations in the *MTHFR* and *PAI-I* genes and thrombosis in patients with COVID-19. However, if we analyze the number of thromboses in patients with at least one mutation that is a marker of thrombophilia, we produce a reliable result (Table 5).

No statistically significant dependence on mutations in inherited thrombophilia studied genes was found when analyzing such parameters as gender, the patient’s admission to the ICU, the degree of lung injury (CTE), and the outcome of hospitalization.

## 4. Discussion

Hereditary thrombophilia is a significant risk factor for thrombosis. The aim of our study was to investigate genetic factors of thrombosis in elderly patients with COVID-19 to determine genetic predictors leading to severe course and thrombotic complications in COVID-19 disease. Two of the mutations studied are the so-called classical inherited thrombophilia mutations: mutations in *FV* Leiden (rs6025) and the prothrombin gene G20210A variant (rs1799963). The term “classical inherited thrombophilia” includes polymorphisms in five genes: Leiden mutations in *FV* gene (rs6025), the prothrombin G20210A variant (rs1799963), protein C, protein S, and antithrombin deficiencies. In 2022, the first genetic population-based study was released that included a total of 29387 elderly individuals [16]. The authors have analyzed the prevalence of each of these five classic thrombophilia factors and their association with thrombosis events in the elderly individuals. Concerning the first two classical thrombophilia genes, the frequencies of *FV* Leiden (rs6025) and the prothrombin G20210A (rs1799963) were completely the same in the large cohort in the cited article and in the cohort of 176 people in our study. In our study we did not estimate the levels of Antithrombin III, Protein C, and Protein S by immunological and coagulological methods since we have dealt with acute thrombosis triggered by COVID-19. American and European guidance [17,18] strongly recommend against the measurement of these proteins in the acute period of thrombosis, because there can be significant variations in Antithrombin III, Protein C, and Protein S levels associated with the consumption of these factors. Testing during this period is inadvisable, because it is impossible to interpret the results adequately. In addition, anticoagulant therapy has a significant impact on the test results. As for the genetic tests for congenital Antithrombin III, Protein C, and Protein S deficiencies our sample cohort will not allow us to obtain reliable results due to the low occurrence of gene mutations leading to the deficiency in appropriate proteins. However, we added four more common mutations to our study, three of which are proven markers of thrombophilia—C677T and A1298C in the *MTHFR* gene, and (-675) 4G/5G in the *PAI-I* gene. The role of the fourth mutation, C494T in the *FII* gene, is debatable (we will discuss this below). The results of our study showed that elderly patients with thrombosis were 1.5 times more likely to be admitted to the ICU than patients of the same age without thrombosis who had other complications, and hospital mortality of the patients with thrombosis was 2.5 times higher.

The works devoted to the influence of hereditary thrombophilia markers on the occurrence of thrombosis in COVID-19 are contradictory. The studies [19,20,21] showed the influence of genes of hereditary thrombophilia on the risk of thrombotic complications in COVID-19; in particular, the highest percentage of genetic mutations was in patients with PE. In other studies, conversely, no such correlations were found [22,23,24]. Probably, the difference in the results in the works can be explained by the fact that they were made at different times and in different places, and the protocol of managing patients with COVID-19 has changed over time. As early as 2020, the leading thrombosis specialists of the International Thrombosis and Hemostasis Society made strong recommendations to administer standard prophylactic or higher doses of heparin in all hospitalized patients with COVID-19 from the moment of admission, if they have no contraindications to these drugs, to control excessive blood clotting [25]. Thus, whereas at the beginning of the pandemic the percentage of venous thrombosis approached 20%, at the end of 2022 the risk of venous thrombosis with adequate therapy dropped to 5.7–8.2% [26]. The results are associated with improved antithrombotic therapy in patients in 2021–2022: timely prescription of anticoagulant therapy and proper dosage choice.

Our research was carried out during the COVID-19 wave caused by the “delta” SARS-CoV-2 strain, when anticoagulant therapy was given to all patients immediately after the diagnosis. Most of the patients (170 of 176; 96.6%) received individual anticoagulant therapy with heparin, low-molecular-weight heparin, or anticoagulants such as rivaroxaban or apixaban. We did not reveal any statistically significant difference between thrombosis and thrombophilia genetic risk factors tested (G1691A mutations in *FV*, G20210A in *FII*, (-675)4G/5G in *PAI-1*, C677T, and A1298C in *MTHFR*). However, there is a tendency for more frequent thrombosis in carriers of G1691A mutations in the *FV* gene and G20210A mutations in the *FII* gene (by 3 and 5 times, respectively). Since these mutations are rare (about 5% and 2% of population, respectively), final conclusions should be tested on larger cohorts of patients. However, in the case of *FII* prothrombin gene mutation C494T (Thr165Met), the thrombosis complications were more frequent (in our case, in 31.2% of patients). We obtained a reliable result showing thrombosis events increase in the carriers of this mutation. In the group without thrombosis, this mutation was found in 27.2% of patients, while in the group with thrombosis, it was found in 43.2% of patients. Prothrombin is converted to thrombin by proteolytic reactions when it initiates the clotting process. Thrombin is a multifunctional serine protease, which is one of the main physiological regulators. Its unique feature is its ability to exhibit both pro- and anticoagulant properties. In carriers of the C494T (Thr165Met) mutation, a substitution of the cytosine C nucleotide in position 494 located on exon 6 of the *FII* gene results in a substitution of the amino acid threonine (Thr) for methionine (Met) in position 165 of the protein. Exon 6 encodes a functionally important amino acid part of prothrombin, called kringle, which is important for protein–protein interactions with clotting factors [27,28]. Changes in the amino acid sequence of the functional center can affect the interaction of thrombin with other proteins in the coagulant and anticoagulant links of hemostasis. One of such factors is Antithrombin III, which inhibits thrombin [29].

There are few studies concerning Thr165Met published so far. However, authors report that it is a genetic marker of thrombophilia, and even in heterozygous form it is also associated with the risk for venous thrombosis [30,31]. These works report on family thrombosis in patients who are both homo- and heterozygous for the Thr165Met mutation. Therefore, we considered both hetero- and homozygous forms of this mutation as a genetic marker of thrombophilia.

Previously, a study [32] showed that the adjusted warfarin dose was higher among patients with the *FII* Thr/Thr genotype (4.2 mg) than among patients with the Thr165Met allele (2.9 mg) [32]. These data, as well as the data from our study, reliably prove the effect of the presence of the Thr165Met allele on the hemostatic system toward thrombosis, which was evident in patients with COVID-19. It can be assumed that the change in the amino acid sequence in the region of the functional center affects the interaction of thrombin with Antithrombin III. Therefore, it partially prevents the inactivation of thrombin, since the anticoagulant effect of heparins is carried out through Antithrombin III. This mutation prevents the eventual full interaction of these proteins and may cause thrombosis even in the presence of anticoagulants. Therefore, we possibly found a significant association between the presence of the Thr165Met mutation and the occurrence of thrombosis in patients with COVID-19, despite taking anticoagulants. Further studies are needed to properly select anticoagulant therapy in patients with COVID-19 who have this mutation.

When analyzing the severity of the course of COVID-19 in multidisciplinary hospital patients, we found no data indicating the influence of genetic mutations in thrombophilia genes on such parameters as patient admission to ICU, CT3-4, and hospital mortality.

A limitation of this study is the relatively small size of the study group (176 people), and the formation of a small group with thrombotic complications. This allows us to consider the results obtained as preliminary, requiring verification in larger studies.

## 5. Conclusions

Thrombosis is an extremely dangerous complication in elderly patients with COVID-19. Of 176 geriatric inpatients (age 50–100 years), 44 (25%) patients had various thrombotic complications, and 72.7% of them were in the intensive care unit. In our study, patients with thrombosis more than 1.5 times more often required NICU treatment than patients without thrombosis with other complications, and their hospital mortality was 2.5 times higher.

In the studied cohort of 176 patients, we did not obtain a reliable result indicating a higher risk of thrombotic complications when taking therapeutic doses of anticoagulants in carriers of the genetic markers for thrombophilia, including G1691A in the *FV* gene, C677T and A1298C in the *MTHFR* gene, G20210A in the *FII* gene, and (-675) 4G/5G in the *PAI-I* gene for COVID-19 disease relative to patients without these markers. However, there was still a pronounced tendency to a higher incidence of thrombosis in patients with markers of hereditary thrombophilia, such as *FV* G1691A and *FII* G20210A mutations. The presence of the C494T (Thr165Met) allele in the *FII* gene in this group of patients showed a statistically significant effect of the mutation on the risk of thrombotic complications despite anticoagulant therapy.

The presence of mutations in the genes of hereditary thrombophilia did not affect such parameters as the severity of the course of COVID-19, the volume of lung damage according to CT scan, the patient’s ICU stay, and hospital mortality as soon as anticoagulant therapy applied. This confirms the need for continued use of the standard anticoagulant therapy in older and older patients.

Thus, our results suggest that even when using therapeutic doses of anticoagulants in elderly patients with COVID-19, the increased risk of thrombotic complications in viral infections in carriers of G1691A mutations in the *FV* gene, and G20210A and C494T in the *FII* gene should be considered, which requires increased coagulogram monitoring and intensified methods of treatment correction.

## Figures and Tables

**Table 1 genes-14-00644-t001:** Mutation points, primers, and fluorescent Taqman probes.

Target Gene	Nucleotide Mutation/ Amino Acid Substitution/ NCBI SNP	Primer/ Probe	Sequence (5′ to 3′)
*FV*	G1691A Arg506Gln rs6025	Forward Reverse w Reverse mt Probe	GACATCGCCTCTGGGCTA CAAGGACAAAATACCTGTATTCCAC CAAGGACAAAATACCTGTATTCCAT (FAM)GCCTGTCCAGGGAT(BHQ1)CTGCTCTTAC
*FII*	G20210A -- rs1799963	Forward Reverse w Reverse mt Probe	TGGAACCAATCCCGTGAAAGAA ACTGGGAGCATTGAGGATC-3 ACTGGGAGCATTGAGGATT (ROX)GAGAGTCACTTTTATTGGGAACCATAG(BHQ2)
*FII*	C494T Thr165Met rs5896	Forward w Forward mt Reverse Probe	ACCCCGACAGCAGCACCTC ACCCCGACAGCAGCACCTT AGCTTACCACAGACAGGGATG (Cy5)GTGCTACACTACAGACCCCACCGTGA(RTQ2)
*MTHFR*	C677T Ala222Va rs1801133	Forward w Forward mt Reverse Probe	GAGAAGGTGTCTGCGGGATC GAGAAGGTGTCTGCGGGATT CATGCCTTCACAAAGCGGAAG (R&G)GATTTCATCATCACGCAGCTTTTCTTTGAGGCTG(BQH2)
*MTHFR*	A1298C Glu429Ala rs1801131	Forward w Forward mt Reverse Probe	GGAGGAGCTGACCAGTGAACA-3’ GAGGAGCTGACCAGTGAACC-3’ GTGACCATTCCGGTTTGGTTCT-3′ (ROX)-GTCTTTGAAGTCTTCGTTCTTTACCTCTCGGGAG(BQH2)
*PAI-I*	(–675) 5G > 4G -- rs1799889	Forward w Forward mt Reverse Probe	AGTCTGGACACGTGGGTG AGTCTGGACACGTGGGTA-3’ CAGCCACGTGATTGTCTAGG-3’ (Cy5)AGCCGTGTATCATCGGAGGCGG(BHQ2)

**Table 2 genes-14-00644-t002:** Characteristics of patients.

Gender, n (%)	Average Age, Years	Vaccinated, n (%)	Assessment of the Degree of Lung Damage on CT Scan n (%)		Thrombotic Complications, (%)		Total Thrombosis, n (%)
Female 112 (63.7) Male, 64 (36.3)	73	52 (29.6)	CT1 CT2 CT3 CT4	51 (30.0) 63 (36.8) 44 (25.0) 18 (10.2)	VTE PE MI ACE	41 (23.7) 5 (2.9) 2 (1.2) 1 (0.58)	44 (25.0), of which 5 (11.4) had combined thromboses

**Table 3 genes-14-00644-t003:** Comparative characteristics of patients with and without thrombotic complications and their relation to hospital mortality and ICU stay.

Characteristics of Patients	Thrombosis Happened (44) n (%)	No Thrombosis (132), n (%)	Estimation of Interrelationship Parameters
Average age	72	72.5	0.89
Female, n (%) Male, n (%)	28 (63.6)	84 (63.6)	OR = 1.0 (CI 95% 0.49–2.03)
	16 (36.4)	48 (36.4)	*p* = 1.0
Hospitalization period (days)	18.5 (4–66)	11 (3–55)	*p* = 0.0001
ICU, n(%)	32 (72.73)	59 (44.7)	OR = 3.29 (CI 95% 1.56–6.96)
			*p* = 0.0013
Hospital mortality	21 (47.73)	25 (18.94)	OR = 3.87 (CI 95% 1.86–8.07)
			*p* = 0.0002

**Table 4 genes-14-00644-t004:** Prevalence of thrombosis events in COVID-19 patients having various genetic thrombophilia markers.

Target Gene/ Mutation	Genotype	Total	Thrombosis Happened, n (%)	No Thrombosis, n (%)	Estimation of Interrelationship Parameters
*FV* G1691A	no mutation	168	40 (90.9)	128 (97.0)	*p* = 0.09 OR = 3.2 (CI 95% 0.76–13.38)
	heterozygote	8	4 (9.1)	4 (3.0)	
*FII* G20210A	No mutation	173	42 (95.5)	131(99.2)	*p* = 0.09 OR = 6.24 (CI 95% 0.55–70.54)
	heterozygote	3	2 (4.5)	1 (0.8)	
*FII* C494	no mutation	121	25 (56.8)	96 (72.7)	*p* = 0.048 OR = 2.03 (CI 95% 0.99–4.12)
	heterozygote + homozygote	55	19 (43.2)	36 (27.2)	
*MTHFR* C677T	no mutation + heterozygote	156	38 (86.4)	118 (89.4)	*p* = 0.58 OR = 1.33 (CI 95% 0.48–3.70)
	homozygote	20	6 (13.6)	14 (10.6)	
*MTHFR * A1298C	no mutation + heterozygote	165	7 (4.2)	125 (95,8)	*p* = 0.37 OR = 1.79 (CI 95% 0.49–6.42)
	homozygote	11	4 (36.4)	7 (63.6)	
*PAI-I* (–675) 5G/4G	no mutation + heterozygote	119	28 (63.6)	91 (68.9)	*p* = 0.52 OR = 1.26 (CI 95% 0.61–2.60)
	homozygote	57	16 (36.4)	41 (31.1)	

Color shows the type of mutation that is a marker of thrombophilia. In some cases it is heterozygous and homozygous mutation, in others it is only homozygous.

**Table 5 genes-14-00644-t005:** Prevalence of thrombosis events in COVID-19 patients having at least one genetic thrombophilia marker and without markers.

Absence/Presence of Genetic Risk Factors Being Studied	Thrombosis Happened/n (%)	No Thrombosis, n (%)	Estimation of Interrelationship Parameters
No genetic risk factors	16 (36.4)	72 (54.5)	*p* = 0.037 OR = 2.10 (CI 95% 1.04–4.24)
Single or more genetic risk factor	28 (63.6)	60(45.5)	

## Data Availability

Data are available on request from the corresponding author.

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
