# Peer review of "Role of Genetic Thrombophilia Markers in Thrombosis Events in Elderly Patients with COVID-19"

_genes, 2023, doi:10.3390/genes14030644_

Round 1

Reviewer 1 Report

Dear authors, when I meet a mistake in the title of an article, it shows me a red flag!

You duplicated “patients” word in a two line tile and none of 12 co-authors sense it.

The authors try to find the prevalence of most common mutations in elderly patients with COVID-19, with or without thrombotic event.

The authors try to find the influence of COVID-19 on thrombotic events, so control groups for this study is not selected properly. If they want to find the effect of thrombophilia markers in COVID-19 patients, they need healthy (or at least thrombosis free) individuals to compare with. 

Author Response

Response to Reviewer 1

  1. Dear authors, when I meet a mistake in the title of an article, it shows me a red flag! You duplicated “patients” word in a two line tile and none of 12 co-authors sense it.

Dear sir/madam,

Thank you for your time in reviewing our work and for your careful attention to it. Of course you're right - it's a shame to make a mistake in the title of an article signed by twelve co-authors. As they say: Too many cooks spoil the broth. We have corrected it.

2  The authors try to find the prevalence of most common mutations in elderly patients with COVID-19, with or without thrombotic event. The authors try to find the influence of COVID-19 on thrombotic events, so control groups for this study is not selected properly. If they want to find the effect of thrombophilia markers in COVID-19 patients, they need healthy (or at least thrombosis free) individuals to compare with,

In this work we compared the prevalence of six genetic thrombophilia mutations in two groups of elderly and old patients with COVID 19: those who suffered thrombosis in the development of the disease and those who did not. Let's clarify what is meant by healthy control? Is it elderly people without thrombosis and without COVID-19 disease?  We have such results. We have the results on the prevalence of these mutations in the same age group without thrombosis before the pandemic: we monitored 183 patients without thrombosis of the mentioned geriatric hospital in 2018. For example, the incidence of the Leiden mutation (FV) in these elderly patients without thrombosis was 6.01%, FII G20210A- 3.9%, FII C494T- 28.8%. But this was before the onset of the pandemic. We considered that it is incorrect to insert these results into the article, because in two years many of them will have COVID-19 disease and will be in either the thrombotic or the thrombosis-free group. And the elderly people who did not get COVID-19 disease in the pandemic are a special and very interesting group, but they also cannot be a control for this study.

Therefore, we have not inserted the results of detecting mutations in healthy (or at least thrombosis free) individuals obtained before the pandemic, even though we have them.

Reviewer 2 Report

thanks for this nice work

some issues need to be rewritten for clarfication of the data

English language should be revised 

Author Response

Response to Reviewer 2

Comments

1. thanks for this nice work

some issues need to be rewritten for clarification of the data

English language should be revised 

Dear Sir/Madam.

Thank you for your encouraging appraisal of our work and your tremendous hard work on improving the English of the article. We tried to fix and improve everything according to your recommendations. Our corrections are in the file attached to the Reply to the Review Report.

Sincerely yours, authors

Reviewer 3 Report

The article entitled “Role of genetic thrombophilia markers in thrombosis events in elderly and old patients with COVID-19” addresses an important clinical topic such as the impact of genetic thrombophilia, such as G1691A FV gene, C677T/ A1298C MTHFR gene, G20210/C494T FII gene, and 4G/5G PAI gene,  on the thrombosis in 176 elderly and old patients with COVID-19. On this basis, the article might be interesting. Hovewer, the authors do not include the evaluation of the natural anticoagulants such as Antithrombin III, Protein C and Protein S. It  is known that a exhaustive study about the ereditary thrombophilia must also consider the genetic deficit of these anticoagulant proteins. Therefore, I suggest to the authors to add this information in their study. Therefore, I think that this article is not suitable for publication in its current version.

Author Response

Response to Reviewer 3

Comments

The article entitled “Role of genetic thrombophilia markers in thrombosis events in elderly and old patients with COVID-19” addresses an important clinical topic such as the impact of genetic thrombophilia, such as G1691A FV gene, C677T/ A1298C MTHFR gene, G20210/C494T FII gene, and 4G/5G PAI gene,  on the thrombosis in 176 elderly and old patients with COVID-19. On this basis, the article might be interesting. Hovewer, the authors do not include the evaluation of the natural anticoagulants such as Antithrombin III, Protein C and Protein S. It  is known that a exhaustive study about the ereditary thrombophilia must also consider the genetic deficit of these anticoagulant proteins. Therefore, I suggest to the authors to add this information in their study. 

Response

Dear sir/madam

Thank you for paying attention to our work and for your review. You are correct, the term "classical thrombophilia" combines the Leiden mutation, the G20210A mutation of the prothrombin gene, deficiencies of antithrombin 3, protein C and protein S. Thrombophilia is an inherited or acquired condition characterized by an excessive tendency of the body to form blood clots. Our work is devoted to the question of the role of inherited markers of thrombophilia in the occurrence of acute thrombosis in COVID19 disease in elderly people. The levels of Antithrombin III, Protein C and Protein S could be detected by immunological and coagulological methods, but these techniques do not reveal the nature of the deficiency - acquired or genetic. Furthermore, the Guidance for the evaluation and treatment of hereditary and acquired thrombophilia [1] and the ESC Guidelines for the diagnosis and management of acute pulmonary embolism developed in collaboration with the European Respiratory Society (ERS)[2] strongly recommend against the measurement of these proteins in the acute period of thrombosis, as there can be significant variations in Antithrombin III, Protein C and Protein S levels associated with consumption of these factors. Testing during this period is inadvisable, because it is impossible to interpret the results adequately. In addition, anticoagulant therapy has a significant impact on the test results.

 Further investigations of genetic markers were performed on DNA isolated from buccal epithelium samples of these patients. We considered the 6 most frequent mutations that are markers of thrombophilia, some of them are not classics, but they are common. The frequency of each of them varies, but exceeds 2% in the European population. The frequency of mutations leading to a genetically determined deficiency of Antithrombin III, Protein C and Protein S is 0.1-0.4% in the European population. Since the cohort of patients under study was 176, the results would have been intentionally unreliable.  In addition, more than 200 mutations leading to deficiency of these proteins have been described, and multiplex PCR, which we use, is not applicable here. Special diagnostic chips are needed to detect these genetic mutations. But, as mentioned above, our sample cohort will not allow us to obtain reliable results due to the low occurrence mutations of the protein genes. 

  However, the frozen DNA collection of COVID-19 patients  is stored in our archive, and we will continue to study the role of gene mutations that aggravate the course of acute respiratory diseases in elderly patients.

  1. Stevens SM, Woller SC, Bauer KA, et al. Guidance for the evaluation and treatment of hereditary and acquired thrombophilia. J Thromb Thrombolysis. 2016;41(1):154-64.
  2. Konstantinides SV, Meyer G, Becattini C, et al. 2019 ESC Guidelines for the diagnosis and management of acute pulmonary embolism developed in collaboration with the European Respiratory Society (ERS). Eur Heart J. 2020;41(4):543-603. 

Round 2

Reviewer 1 Report

The manuscript is improved.

Reviewer 3 Report

In the revised version of the article entitled “Role of genetic thrombophilia markers in thrombosis events in elderly and old patients with COVID-19” the authors modified the manuscript in according to the comments of the reviewer. Therefore, I think that the article is suitable for publication in its current revised version.